# Diagnostic Value of IP-10 Level in Plasma and Bronchoalveolar Lavage Fluid in Children with Tuberculosis and Other Lung Diseases

**DOI:** 10.3390/diagnostics12040840

**Published:** 2022-03-29

**Authors:** Agnieszka Strzelak, Anna Komorowska-Piotrowska, Katarzyna Krenke, Wioletta Zagórska, Witold Bartosiewicz, Wojciech Feleszko, Marek Kulus

**Affiliations:** Department of Pediatric Pulmonology and Allergy, Medical University of Warsaw, 02-091 Warsaw, Poland; agnieszka.strzelak@wum.edu.pl (A.S.); katarzyna.krenke@wum.edu.pl (K.K.); wioletta.zagorska@wum.edu.pl (W.Z.); witold.bartosiewicz@wum.edu.pl (W.B.); wojciech.feleszko@wum.edu.pl (W.F.); marek.kulus@wum.edu.pl (M.K.)

**Keywords:** child, CXCL10, bronchoscopy, biomarker, latent tuberculosis infection

## Abstract

Objectives: IP-10 has been proposed as a new diagnostic biomarker for *Mycobacterium tuberculosis* infection (MTBI). However, data on IP-10 concentration in bronchoalveolar lavage fluid (BALF) for pediatric tuberculosis are lacking. Aim: To determine IP-10 levels in unstimulated BALF and plasma in children with and without MTBI. Methods: IP-10 concentrations in BALF and plasma were measured in children hospitalized with suspected tuberculosis or other respiratory disease and scheduled for bronchoscopy. Thirty-five children were enrolled: 13 with suspected tuberculosis and 22 controls. The association between IP-10 and age was examined. Results: The IP-10 expression was increased in BALF compared to plasma (*p* = 0.008). We noticed higher BALF IP-10 levels in children with asthma, interstitial lung disease, and lung anomaly than in children with MTBI and other respiratory tract infections, but the differences were statistically insignificant. There was a moderate correlation between plasma and BALF IP-10 concentrations (r_s_ = 0.46, *p* = 0.018). No correlation between IP-10 level and age was detected. Conclusions: IP-10 is detectable in unstimulated BALF in children with respiratory diseases, reaches higher concentrations in unstimulated BALF vs plasma, and does not correlate with age. However, it could not discriminate MTBI from other respiratory diseases.

## 1. Introduction

Despite many efforts, tuberculosis (TB) remains one of the leading causes of mortality globally. According to the World Health Organization (WHO), children accounted for 11% of all TB cases in 2020 [1]. However, childhood TB has long been neglected, as children are considered not to take part in the transmission of the disease and therefore this had not been a public health priority [2]. Additionally, the diagnosis of childhood TB is still very challenging, despite significant advances in diagnostic technologies and techniques that have emerged over the last twenty years. This results from uncharacteristic clinical presentation, difficulties in obtaining samples for microbiological evaluation, and paucibacillary disease [1,3].

Furthermore, the performance of diagnostic tests based on delayed-type hypersensitivity reactions, such as the tuberculin skin test (TST) and interferon-γ (IFN-γ) release assays (IGRAs), also has considerable limitations in children. TST is operator-dependent, has limited specificity, and can produce false-positive results in children vaccinated with BCG, retested with TST, or exposed to environmental mycobacteria [4,5]. While the specificity of IGRAs is considered higher than that of TST, their performance in children is still a subject of debate [6,7,8]. At the same time, children are at a higher risk of progression to active TB following infection and to develop more severe forms of the disease [9]. Clearly, there is a significant and pressing need for new, easy to use, cost-effective, and accurate diagnostic tools for paucibacillary TB in the young.

One of the most excessively studied potential new markers of *Mycobacterium tuberculosis* (Mtb.) infection is interferon-γ (IFN-γ) inducible protein of 10 kDa (IP-10). IP-10 is a chemokine secreted primarily by the antigen-presenting cells upon T-cell stimulation that navigates activated T-cells to the foci of inflammation [10]. IP-10 is expressed in much higher quantities than IFN-γ. Therefore, it has the potential to improve the performance characteristics of Quantiferon-TB Gold In-Tube (QFT) in particular in human immunodeficiency virus (HIV)-infected individuals and young children [11,12,13,14]. Tests based on plasma IP-10 measurements have been shown to perform comparably to QFT in the majority of studied populations [15]. Apart from blood, an increased expression of IP-10 in adult TB patients has been detected in urine, lymph node aspirates, lung tuberculous granulomas, pleural effusion, and bronchoalveolar lavage fluid (BALF) [10,16,17,18,19,20,21,22]. In children, data on IP-10 concentration in specimens other than blood are limited, with only one study examining IP-10 in urine [23]. To date, to the best of our knowledge, there is no published data on the utility of IP-10 measurement in BALF in children with Mtb. infection.

The aim of the present study was to examine the level of IP-10 in unstimulated BALF and plasma in children infected with *Mycobacterium tuberculosis* and children with other respiratory diseases from a low TB-endemic country.

## 2. Materials and Methods

### 2.1. Patients

Children were enrolled at the Department of Pediatric Pneumonology and Allergy at the Medical University of Warsaw, Poland, from June 2012 to July 2015. Eligibility criteria were age <18 years, clinical suspicion of TB, and a planned bronchoscopy. Exclusion criteria included known immunosuppression, including HIV infection, malignancy, exacerbation of a chronic disease, previous antituberculous treatment, and a TST performed within the last 18 months to minimize the risk of boosting. The control group consisted of children with other respiratory diseases who were scheduled for an elective bronchoscopy in our department as a part of the routine diagnostic workup. Exclusion criteria encompassed known immunosuppression, including HIV infection, malignancy, exacerbation of a chronic disease, and a known TB contact. The study protocol was approved by the Medical University of Warsaw Ethics Committee and informed consent was obtained from the child’s parent or legal guardian before enrollment. Demographic and clinical data, including history of BCG vaccination, peripheral blood differential count, and chest radiographic examination results were recorded. Children suspected of TB underwent TST and/or QFT tests as a routine diagnostic procedure. Bacteriological evaluation including microscopy, molecular microbiological tests, and culture was performed in all children suspected of active TB. The decision to perform bronchofiberoscopy with bronchoalveolar lavage (BAL) was made by leading physicians after initial TB workup in children with a high risk of infection, e.g., household contact, positive immunological test results, clinical or radiological signs and symptoms suggestive of TB. The diagnosis of active TB and latent tuberculous infection (LTBI) was based on clinical, radiological, and microbiological data. Active TB was diagnosed in symptomatic patients with microbiologically confirmed TB (positive microscopy, PCR or culture result), or in patients fulfilling at least two of the following three criteria: (1) clinical symptoms suggestive of TB, (2) radiological findings consistent with TB, and (3) a history of contact with pulmonary TB; all in conjunction with a positive clinical response to antituberculous treatment. LTBI was defined as a positive TST and/or IGRA result in children with a history of contact with pulmonary TB, a normal chest X-ray, and neither clinical symptoms nor microbiological confirmation of TB. Asymptomatic children with a history of contact with pulmonary TB, not fulfilling the criteria for ATB or LTBI, were classified as TB contacts.

### 2.2. Collection and Processing of BAL and Blood Samples

BALF samples were collected with a flexible video bronchoscope (Pentax, Tokyo, Japan). All bronchoscopies were performed in the Department of Pediatric Pneumonology and Allergy, Medical University of Warsaw, Poland, and according to the European and Polish guidelines [24,25,26]. In brief, following intravenous sedation at the anaesthesiologist’s discretion and local upper airway anesthesia with lidocaine (1%), the bronchoscope was gently wedged in the lobe with a radiological abnormality or, if absent, in the right middle lobe or in the left lung lingula. BAL was performed by instillation and subsequent withdrawal of up to five 20-mL aliquots of sterile pre-warmed normal saline according to the weight of a child (approximately 3 mL/kg of BAL volume). The recovered fluid was divided into three separate containers. Fluid recovery of at least 40% of instilled BAL volume was considered acceptable. The first sample was sent for microbiological analysis, and the second and the third samples were placed on ice for immediate transport to the laboratory. BALF samples were mixed, filtered, and immediately centrifuged. A total cell count, cell viability (trypan blue exclusion test), and differential cell count of BALF leukocytes were determined by light microscopy. The cell-free BALF supernatant was frozen at −70 °C until further analysis. Children with Mtb. infection underwent bronchoscopy before the onset of antituberculous treatment. Serum for IP-10 measurement was obtained from whole blood drawn on the same day as the BAL using standard procedures.

### 2.3. Determination of IP-10 Concentration

Samples were thawed and IP-10 expression was analyzed in undiluted BALF supernatants and plasma using the Quantikine ELISA Human IP-10 Immunoassay (R&D Systems Europe, Ltd., Abingdon, UK) according to the manufacturer’s recommendations. All samples were measured in duplicate. The concentration range of detection was 7.8–500 pg/mL. IP-10 concentrations are expressed in pg/mL.

### 2.4. Statistics

Statistical analysis was performed using STATISTICA 13.1 (SatSoft). Figures were prepared in GraphPad Prism version 8 (GraphPad Software, San Diego, CA, USA). Data were presented as medians and interquartile ranges (IQR). Non-parametric tests were applied due to the non-normal distribution of variables (Kolmogorov-Smirnov normality test). We performed the chi-square test or Fisher’s exact test for qualitative variables and the Kruskall-Wallis test and the Mann Whitney test were used for quantitative variables. Plasma and BALF IP-10 concentrations in paired samples were compared with the Wilcoxon signed rank test. Correlation analysis was performed to assess biomarker correlation with clinical variables with the Spearman’s rank test. Statistical significance was considered for *p* values of <0.05.

## 3. Results

Out of thirty-nine eligible children who underwent a bronchofiberoscopy with BAL in our department during the study period, thirty-five children were included in this study, thirteen children to the study group, and twenty-two to the control group. The remaining three (8%) children were excluded due to blood contamination (*n* = 1), hemolysis (*n* = 1), and lab handling error (*n* = 1). The children’s median age was six years (IQR:4–9 years) and twenty-two (63%) of them were females. In the study group, we identified twelve children with Mtb. infection, all of whom had a positive TST result. QFT was performed in eight children from the study group and was positive in seven cases. There were three cases of active TB and nine children with LTBI in the study group. All three children diagnosed with active TB had a history of household contact with pulmonary TB, positive TST, and one boy also had a positive QFT result, two had abnormal chest X-ray results with opacification, atelectasis, and lymphadenopathy with calcifications, and all presented abnormalities in bronchofiberscopy including altered mucous membranes (*n* = 3), presence of secretion (*n* = 3), and narrowing of the bronchial tree (*n* = 2). One five-year-old boy with a history of recent TB contact and negative TST and QFT results was classified as an uninfected TB contact and was excluded from further analysis. The control group comprised five children with asthma, six children with interstitial lung disease (ILD), four children with lung anomalies, and seven children with other respiratory tract infections (RTI). None of children from the control group had a known TB contact. TST and/or QFT were performed in a few cases as a part of differential diagnostic workup. All enrolled children were BCG vaccinated at infancy, and none was diagnosed with HIV infection. A summary of the study group characteristics is presented in Table 1. Children in the study and the control group had comparable baseline characteristics. BALF cytological results were available for twenty-two (63%) participants and are presented in Table 2. Although some differences in BAL cytological characteristics between the groups were noted, statistical relevance was demonstrated only for a higher proportion of lymphocytes in BALF of Mtb.-infected children in comparison with the control group (*p* = 0.02).

### 3.1. BALF and Plasma Concentrations of IP-10

The median IP-10 BALF concentration was 229 pg/mL (IQR 138–483 pg/mL) in Mtb.-infected children and 412 pg/mL (IQR 110.4–851.8 pg/mL) in Mtb.-uninfected controls. Plasma IP-10 concentration was measured in 26 (74.3%) children, among whom eight were Mtb.-infected and 18 were not. The median plasma IP-10 concentration in Mtb.-infected children was 136 pg/mL (IQR 109.3–207 pg/mL) and in the control group it equaled 129 pg/mL (IQR 73.5–206 pg/mL). Although children in the control group had a higher BALF IP-10 level, no significant differences in BALF or plasma IP-10 concentrations were demonstrated between the study and the control group (Figure 1).

Subsequently, we performed a subgroup analysis including subgroups of children in the study and the control group. The median IP-10 BALF concentrations in children with ATB, LTBI, asthma, ILD, lung anomalies, and RTI were 414 pg/mL (IQR 104–598 pg/mL), 191 pg/mL (IQR 157–371 pg/mL), 490 pg/mL (IQR 216–1486 pg/mL), 578 pg/mL (IQR 410.3–852 pg/mL), 571 pg/mL (IQR 90.7–1261.7 pg/mL), and 281.7 pg/mL (IQR 110.4–414 pg/mL), respectively (Figure 2A). The median IP-10 plasma levels in children with ATB, LTBI, asthma, ILD, lung anomalies, and RTI were 82 pg/mL (IQR 50–113 pg/mL), 152 pg/mL (IQR 129–253 pg/mL), 131 pg/mL (IQR 101–331 pg/mL), 186 pg/mL (IQR 126.0–206 pg/mL), 127.8 pg/mL (IQR 84.1–258.1 pg/mL), and 115.1 pg/mL (IQR 72.5–165 pg/mL), respectively (Figure 2B). The BALF IP-10 levels were higher in children with asthma, ILD, and lung anomalies than in children with ATB, LTBI and the RTI group, although the differences were not significant. Median IP-10 plasma levels were the lowest in children with ATB but, similarly, no significant differences in IP-10 plasma concentrations were noticed between the study groups (*p* > 0.05) (Figure 2).

### 3.2. Correlation of BALF and Plasma IP-10 Levels with Age

Next, we compared IP-10 levels in plasma with its concentrations in BALF. Importantly, we demonstrated significantly higher IP-10 concentrations in BALF than those found in plasma (Figure 3). There was a moderate correlation between plasma and BALF IP-10 levels (r_s_ = 0.46, *p* = 0.018). When we examined the impact of child’s age on biomarker concentration in both compartments, we found no correlation between plasma or BALF IP-10 concentration and age (r_s_ = −0.026, *p* = 0.9 and r_s_ = −0.032, *p* = 0.85, respectively). IP-10 in BALF did not correlate with BALF recovery or BALF cytology results and, similarly, plasma IP-10 did not correlate with blood differential count results (data not shown).

## 4. Discussion

In the present study performed in a low TB-endemic country with a highly BCG vaccinated population, we demonstrated that IP-10 is detectable in the BALF of children with *Mycobacterium tuberculosis* infection and other respiratory diseases. Importantly, we showed significantly higher concentrations of IP-10 in BALF than those found in the plasma of children with respiratory diseases. We demonstrated that BALF IP-10 is significantly correlated with plasma IP-10 but it is not associated with age. These findings indicate that IP-10 measurement in BALF has the potential of becoming an additional biomarker in the diagnosis of respiratory diseases in children.

IP-10 has been proposed as a new TB biomarker for adults and children [15]. Indeed, higher IP-10 levels, albeit inconsistently, have been demonstrated in sera of TB patients and subjects latently infected with *Mycobacterium tuberculosis* compared with uninfected controls [12,13,14,27,28,29]. Similar observations were made in the studies assessing other biological specimens obtained from Mtb.-infected adults and the urine of children with active TB [10,16,17,18,23]. The observed conflicting results of some published studies have been attributed to the differences in the studies’ designs (local TB endemicity, populations studied, and methods of IP-10 concentration determination). While a considerable number of studies employed QFT test tubes to measure IP-10 concentration, the cost of this method remains prohibitive in resource-poor settings. Several reports demonstrated increased concentrations of IP-10 in unstimulated plasma and urine of subjects with ATB compared with adults and children without active disease [23,30,31,32,33]. Here we confirm that IP-10 is detectable also in unstimulated BALF of children with Mtb. infection and other respiratory diseases, possibly reflecting the local pro-inflammatory cellular milieu in childhood respiratory diseases. In agreement with our findings, IP-10 has been previously related to infections with other pathogens and conditions associated with chronic inflammation [15,34,35].

There are few studies investigating IP-10 in BALF samples published to date. In adults, higher IP-10 concentrations were observed, among others, in patients with TB, asthma, sarcoidosis, acute pulmonary exacerbations in cystic fibrosis, and in acute rejection after lung transplantation [36,37,38]. An increased expression of IP-10 has been shown in BAL cells and fluid of TB patients. Sauty et al. demonstrated that IP-10 mRNA was expressed in the bronchial epithelium and that IP-10-positive cells were significantly increased in BALF obtained from TB patients compared with healthy controls [17]. Elevated expression of IP-10 mRNA in BAL cells from TB patients was next confirmed by others [21,36].

Furthermore, the differences in IP-10 levels in BAL specimens seem to reflect the severity of the disease. In patients with active TB with or without HIV-coinfection, higher concentrations of IP-10 in BAL cells supernatants and in BAL fluid have been associated with non-cavitary TB [20,22]. Walrath et al. showed an increased level of IP-10 in BALF also in asymptomatic latently infected subjects [19]. Following in vivo and in vitro stimulation with a purified protein derivative (PPD) of *Mycobacterium tuberculosis,* an augmented expression of this molecule was detected only in BALF and BAL cells obtained from patients with a positive TST result. In our study, however, we did not observe significantly higher IP-10 concentrations in Mtb.-infected individuals when compared with uninfected controls or in active TB patients when compared with the LTBI group. Unlike some previous reports, we did not incubate BAL cells with Mtb.-specific antigens, an approach taken previously by others investigating biomarkers in BALF, blood, and urine [22,23,33]. Nevertheless, we were able to show no correlation between IP-10 and the age of children, which is of great importance in pediatrics. Our results indicate that IP-10 measurement in BALF is not only technically feasible, but carries the potential of becoming an additional TB biomarker in childhood TB regardless of age. Further research should, therefore, incorporate BAL cell incubation with Mtb.-derived peptides and include a larger number of children with different stages of Mtb. infection and other respiratory diseases. Importantly, processing of the bronchoalvelar fluid once BAL is performed is relatively simple, but the information it provides about the pulmonary milieu during the infection with *Mycobacterium tuberculosis* cannot be overestimated.

While our study is the first to evaluate the potential use of IP-10 measurement in BAL fluid obtained from children under investigation for TB, we acknowledge that it bears several limitations. First of all, due to the small sample size, the present study might have lacked statistical power to show significant differences in IP-10 BALF levels between the groups. Although we observed some discrepancies in BAL cytology between the groups, these were too small to reach the level of significance and hampered us from demonstrating associations between BAL cell differential count and IP-10 level. Moreover, for ethical reasons, the control group consisted of children with other respiratory diseases in whom a planned bronchoscopy was a routine diagnostic procedure. Another drawback of the present study lies in the fact that not all of the Mtb.-infected children had an available QFT result. However, all TST positive results were >10 mm, which is in agreement with national and European guidelines concerning BCG-vaccinated individuals [39,40]. While in some children from the control group the TB diagnostics was performed with negative results, Mtb. infection cannot be excluded in the remaining participants. Taken together, this could have caused the potential misclassification of patients. On the other hand, TB-incidence in Poland is low, and a recent TB contact was carefully excluded before enrollment to the control group. In addition, our study population did not include children under the age of 2 years. Therefore, the concentrations of IP-10 in BALF obtained from the youngest children remain unknown.

In conclusion, the results of the present study confirm that IP-10 is detectable in BALF obtained from children, does not correlate with age, and that IP-10 in unstimulated BALF reaches even higher concentrations than in unstimulated plasma in children with respiratory diseases. The evaluation of IP-10 in BALF of children suspected of TB is a novelty, and these findings could provide a framework for a better understanding of disease processes. Since bronchoscopy is a part of the TB diagnostic work-up, the obtained material could serve as an additional sample for immunodiagnostic assays facilitating the accurate diagnosis. Further investigations are certainly needed to evaluate the robustness of these results and to determine the advantage of IP-10 measurement in BALF in children with different stages of Mtb. infection in a larger population.

## Figures and Tables

**Figure 1 diagnostics-12-00840-f001:**
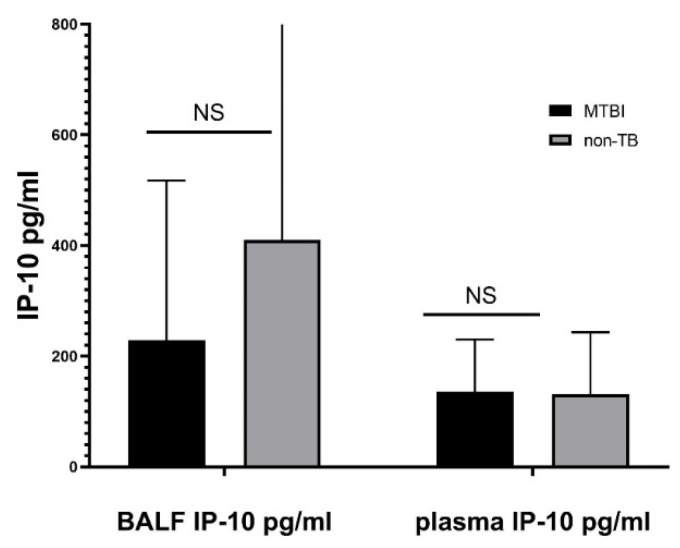
IP-10 responses in BALF and plasma in *Mycobacterium tuberculosis* infected subjects and *Mycobacterium tuberculosis* uninfected controls. Definition of abbreviations: MTBI—children infected with *Mycobacterium tuberculosis*; non-TB—children not infected with *Mycobacterium tuberculosis*, BALF—bronchoalveolar lavage fluid, NS—non-significant.

**Figure 2 diagnostics-12-00840-f002:**
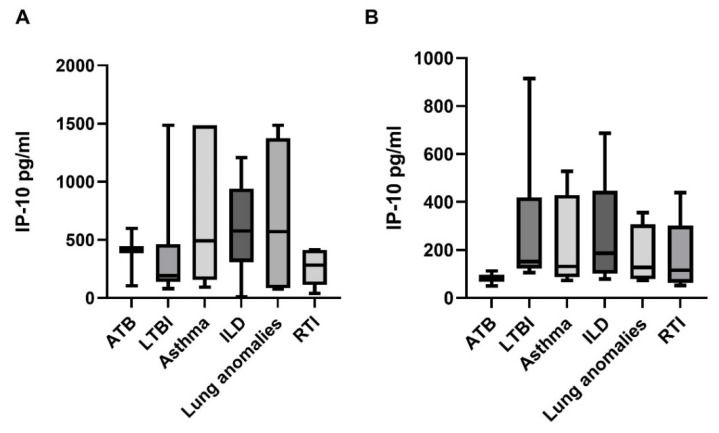
IP-10 responses in bronchoalveolar lavage fluid (**A**) and plasma (**B**) in the study groups. Definition of abbreviations: ATB—active tuberculosis, LTBI—latent tuberculous infection, ILD—interstitial lung disease, RTI—respiratory tract infection.

**Figure 3 diagnostics-12-00840-f003:**
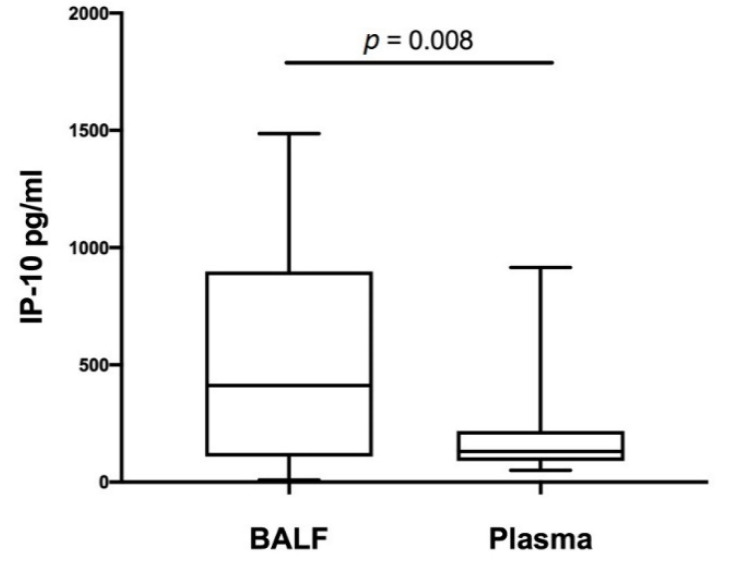
Comparison between IP-10 level in BALF and plasma. Definition of abbreviations: BALF—bronchoalveolar lavage fluid.

**Table 1 diagnostics-12-00840-t001:** Characteristics of the study participants.

	Study Group	Control Group				
	*n* = 13 (%)	Total *n* = 22 (%)	Asthma *n* = 5 (%)	ILD *n* = 6 (%)	Lung Anomalies *n* = 4 (%)	RTI *n* = 7 (%)
Sex						
Females	10 (77)	12 (55)	2 (40)	3 (50)	2 (50)	5 (71)
Males	3 (23)	10 (45)	3 (60)	3 (50)	2 (50)	2 (29)
Age (yrs), median (IQR)	5 (4–6)	7 (4–12)	12 (6–13)	10 (3–12)	6 (4–10.5)	6 (2–9)
≤5 yrs	5 (38)	6 (27)	0 (0)	2 (33)	1 (25)	3 (43)
0–2 yrs	0 (0)	1 (5)	0 (0)	2 (33)	0 (0)	0 (0)
Origin						
Poland	13 (100)	22 (100)	5 (100)	6 (100)	4 (100)	7 (100)
BMI						
3–97 percentile	13 (100)	22 (100)	5 (100)	6 (100)	4 (100)	7 (100)
BCG vaccinated						
Yes	13 (100)	22 (100)	5 (100)	6 (100)	4 (100)	7 (100)
BCG scar						
Present	11 (85)	19 (86)	4 (80)	5 (83)	4 (100)	6 (86)
Not recorded	0 (0)	0 (0)	0 (0)	0 (0)	0 (0)	0 (0)
TST						
TST-positive	12 (92)	0 (0)	0 (0)	0 (0)	0 (0)	0 (0)
TST-negative	1 (8)	5 (23)	1 (20)	0 (0)	1 (25)	3 (43)
TST not performed	0 (0)	17 (77)	4 (80)	6 (100)	3 (75)	4 (57)
QFT						
QFT-positive	7 (54)	0 (0)	0 (0)	0 (0)	0 (0)	0 (0)
QFT-negative	1 (8)	4 (18)	1 (20)	0 (0)	0 (0)	3 (43)
QFT-indeterminate	0 (0)	0 (0)	0 (0)	0 (0)	0 (0)	0 (0)
QFT Not performed	5 (38)	18 (82)	4 (80)	6 (100)	4 (100)	4 (57)
Known TB contact						
Yes	13 (100)	0 (0)	0 (0)	0 (0)	0 (0)	0 (0)
No	0 (0)	22 (100)	5 (100)	6 (100)	4 (100)	7 (100)
Index case						
Smear–positive	10 (77)	n/A	n/A	n/A	n/A	n/A
Smear–negative	1 (8)	n/A	n/A	n/A	n/A	n/A
Not known	2 (15)	n/A	n/A	n/A	n/A	n/A
TB diagnosis						
Microbiological	0 (0)	n/A	n/A	n/A	n/A	n/A
Clinical	3 (100)	n/A	n/A	n/A	n/A	n/A
Clinical symptoms	2 (66)	n/A	n/A	n/A	n/A	n/A
Chest X-ray abnormalities	2 (66)	n/A	n/A	n/A	n/A	n/A
Abnormal bronchofiberoscopy	3 (100)	n/A	n/A	n/A	n/A	n/A
IP-10 measured in BALF	13 (100)	22 (100)	5 (100)	6 (100)	4 (100)	7 (100)
IP-10 measured in plasma	8 (62)	18 (82)	4 (80)	5 (83)	4 (100)	5 (71)

Definition of abbreviations: yrs—years, IQR—interquartile range, BMI—body max index, BCG—*bacille Calmette-Guérin*, TST—tuberculin skin test, QFT—Quantiferon-TB Gold In-Tube, TB—tuberculosis, IP-10—interferon-γ inducible protein of 10 kDa, BALF—bronchoalveolar lavage fluid, ILD—interstitial lung disease, RTI—respiratory tract infection.

**Table 2 diagnostics-12-00840-t002:** Cytological bronchoalveolar lavage data of patients and control subjects (presented as medians and IQRs).

	BALF Recovery (%)	BALF Cell Count, 10^3^/mL	Macrophages (%)	Lymphocytes (%)	Neutrophils (%)	Eosinophils (%)
**Mtb.-infected**	53 (49.4–63)	155 (107–293)	52 (49.4–64)	44 (31.7–45)	4 (3.0–4)	0 (0.2–1)
** ATB**	56 (50–61)	129 (81–177)	64 (36–65)	32 (15–61)	4 (3–20)	0 (0–1)
** LTBI**	50 (49–64)	271 (133–410)	51 (49–52)	45 (44–45)	4 (3–4)	1 (0–2)
**Non-TB**	62 (50–74)	225 (107–355)	72 (54–79)	16 (13–26)	5 (2–17)	0 (0–1)
** Asthma**	59 (54–65)	187.5 (107–290)	78 (61–82)	15 (15–33)	5 (3–7)	0 (0–1)
** ILD**	61 (60.0–74)	341 (131.8–687.5)	73 (66–87)	18 (8–27)	2 (0–5)	0 (0–0)
** Lung anomalies**	73 (60–78)	313 (286.7–340)	40 (24–56)	12 (8–17)	46 (26–66)	2 (1–3)
** RTI**	56.7 (42.2–68)	78 (60–230.7)	71 (52.7–76)	21 (14.3–26)	13 (5.9–16)	0 (0.1–3)

Definition of abbreviations: Mtb.-infected—children infected with *Mycobacterium tuberculosis*, ATB—active tuberculosis, LTBI—latent tuberculous infection, Non-TB—children not infected with *Mycobacterium tuberculosis*, ILD—interstitial lung disease, RTI—respiratory tract infection, BALF—bronchoalveolar lavage fluid.

## Data Availability

The data presented in this study are available on request from the corresponding author.

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
