# Peer review of "Diagnostic Value of IP-10 Level in Plasma and Bronchoalveolar Lavage Fluid in Children with Tuberculosis and Other Lung Diseases"

_diagnostics, 2022, doi:10.3390/diagnostics12040840_

Round 1

Reviewer 1 Report

It is recommended to correct the title of the study.  Diagnostic value of IP-10 in TB has been analyzed. The proposed title – e.g. «Diagnostic value of IP-10 level in children with tuberculosis and other lung diseases»

The results of the study are valuable for TB diagnosis, as currently this problem has not been solved yet. References’ numbers require adjustment in accordance with rules of the journal. The authors should apply the epidemiologic data that is presented in the WHO Report for 2021.

The aim of the study needs re-assessment to make it consistent with design of the study.

Data on the lines 62-77 is not enough clear and may need re-structuring.

Inclusion criteria should be reconsidered: clinical suspicion of TB should not be the inclusion criterion, because TB children often may have no evident clinical symptoms. 

It should be included in exclusion criteria – HIV infection, chronic diseases in exacerbation stage, and oncology diseases.

It is not clear characteristic of children in the control group.

Colleagues should include the data of the Ethical approval of the study and declare the consent of the parents for children to participate.  

The number of patients in the study group (n=13) is not sufficient to perform representative statistical analysis.

Colleagues did not conclude on sensitivity and specificity of IP-10 that would increase scientific value of the study.

Conclusions are not clear, although the study itself is comprehensive and analysis was performed on the high level. Still aim and objective of the study are not fully evident.

It is recommended to correct the title of the study.  Diagnostic value of IP-10 in TB has been analyzed. The proposed title – e.g. «Diagnostic value of IP-10 level in children with tuberculosis and other lung diseases»

The results of the study are valuable for TB diagnosis, as currently this problem has not been solved yet. References’ numbers require adjustment in accordance with rules of the journal. The authors should apply the epidemiologic data that is presented in the WHO Report for 2021.

The aim of the study needs re-assessment to make it consistent with design of the study.

Data on the lines 62-77 is not enough clear and may need re-structuring.

Inclusion criteria should be reconsidered: clinical suspicion of TB should not be the inclusion criterion, because TB children often may have no evident clinical symptoms. 

It should be included in exclusion criteria – HIV infection, chronic diseases in exacerbation stage, and oncology diseases.

It is not clear characteristic of children in the control group.

Colleagues should include the data of the Ethical approval of the study and declare the consent of the parents for children to participate.  

The number of patients in the study group (n=13) is not sufficient to perform representative statistical analysis.

Colleagues did not conclude on sensitivity and specificity of IP-10 that would increase scientific value of the study.

Conclusions are not clear, although the study itself is comprehensive and analysis was performed on the high level. Still aim and objective of the study are not fully evident.

It is recommended to correct the title of the study.  Diagnostic value of IP-10 in TB has been analyzed. The proposed title – e.g. «Diagnostic value of IP-10 level in children with tuberculosis and other lung diseases»

The results of the study are valuable for TB diagnosis, as currently this problem has not been solved yet. References’ numbers require adjustment in accordance with rules of the journal. The authors should apply the epidemiologic data that is presented in the WHO Report for 2021.

The aim of the study needs re-assessment to make it consistent with design of the study.

Data on the lines 62-77 is not enough clear and may need re-structuring.

Inclusion criteria should be reconsidered: clinical suspicion of TB should not be the inclusion criterion, because TB children often may have no evident clinical symptoms. 

It should be included in exclusion criteria – HIV infection, chronic diseases in exacerbation stage, and oncology diseases.

It is not clear characteristic of children in the control group.

Colleagues should include the data of the Ethical approval of the study and declare the consent of the parents for children to participate.  

The number of patients in the study group (n=13) is not sufficient to perform representative statistical analysis.

Colleagues did not conclude on sensitivity and specificity of IP-10 that would increase scientific value of the study.

Conclusions are not clear, although the study itself is comprehensive and analysis was performed on the high level. Still aim and objective of the study are not fully evident.

It is recommended to correct the title of the study.  Diagnostic value of IP-10 in TB has been analyzed. The proposed title – e.g. «Diagnostic value of IP-10 level in children with tuberculosis and other lung diseases»

The results of the study are valuable for TB diagnosis, as currently this problem has not been solved yet. References’ numbers require adjustment in accordance with rules of the journal. The authors should apply the epidemiologic data that is presented in the WHO Report for 2021.

The aim of the study needs re-assessment to make it consistent with design of the study.

Data on the lines 62-77 is not enough clear and may need re-structuring.

Inclusion criteria should be reconsidered: clinical suspicion of TB should not be the inclusion criterion, because TB children often may have no evident clinical symptoms. 

It should be included in exclusion criteria – HIV infection, chronic diseases in exacerbation stage, and oncology diseases.

It is not clear characteristic of children in the control group.

Colleagues should include the data of the Ethical approval of the study and declare the consent of the parents for children to participate.  

The number of patients in the study group (n=13) is not sufficient to perform representative statistical analysis.

Colleagues did not conclude on sensitivity and specificity of IP-10 that would increase scientific value of the study.

Conclusions are not clear, although the study itself is comprehensive and analysis was performed on the high level. Still aim and objective of the study are not fully evident.

It is recommended to correct the title of the study.  Diagnostic value of IP-10 in TB has been analyzed. The proposed title – e.g. «Diagnostic value of IP-10 level in children with tuberculosis and other lung diseases»

The results of the study are valuable for TB diagnosis, as currently this problem has not been solved yet. References’ numbers require adjustment in accordance with rules of the journal. The authors should apply the epidemiologic data that is presented in the WHO Report for 2021.

The aim of the study needs re-assessment to make it consistent with design of the study.

Data on the lines 62-77 is not enough clear and may need re-structuring.

Inclusion criteria should be reconsidered: clinical suspicion of TB should not be the inclusion criterion, because TB children often may have no evident clinical symptoms. 

It should be included in exclusion criteria – HIV infection, chronic diseases in exacerbation stage, and oncology diseases.

It is not clear characteristic of children in the control group.

Colleagues should include the data of the Ethical approval of the study and declare the consent of the parents for children to participate.  

The number of patients in the study group (n=13) is not sufficient to perform representative statistical analysis.

Colleagues did not conclude on sensitivity and specificity of IP-10 that would increase scientific value of the study.

Conclusions are not clear, although the study itself is comprehensive and analysis was performed on the high level. Still aim and objective of the study are not fully evident.

It is recommended to correct the title of the study.  Diagnostic value of IP-10 in TB has been analyzed. The proposed title – e.g. «Diagnostic value of IP-10 level in children with tuberculosis and other lung diseases»

The results of the study are valuable for TB diagnosis, as currently this problem has not been solved yet. References’ numbers require adjustment in accordance with rules of the journal. The authors should apply the epidemiologic data that is presented in the WHO Report for 2021.

The aim of the study needs re-assessment to make it consistent with design of the study.

Data on the lines 62-77 is not enough clear and may need re-structuring.

Inclusion criteria should be reconsidered: clinical suspicion of TB should not be the inclusion criterion, because TB children often may have no evident clinical symptoms. 

It should be included in exclusion criteria – HIV infection, chronic diseases in exacerbation stage, and oncology diseases.

It is not clear characteristic of children in the control group.

Colleagues should include the data of the Ethical approval of the study and declare the consent of the parents for children to participate.  

The number of patients in the study group (n=13) is not sufficient to perform representative statistical analysis.

Colleagues did not conclude on sensitivity and specificity of IP-10 that would increase scientific value of the study.

Conclusions are not clear, although the study itself is comprehensive and analysis was performed on the high level. Still aim and objective of the study are not fully evident.

It is recommended to correct the title of the study.  Diagnostic value of IP-10 in TB has been analyzed. The proposed title – e.g. «Diagnostic value of IP-10 level in children with tuberculosis and other lung diseases»

The results of the study are valuable for TB diagnosis, as currently this problem has not been solved yet. References’ numbers require adjustment in accordance with rules of the journal. The authors should apply the epidemiologic data that is presented in the WHO Report for 2021.

The aim of the study needs re-assessment to make it consistent with design of the study.

Data on the lines 62-77 is not enough clear and may need re-structuring.

Inclusion criteria should be reconsidered: clinical suspicion of TB should not be the inclusion criterion, because TB children often may have no evident clinical symptoms. 

It should be included in exclusion criteria – HIV infection, chronic diseases in exacerbation stage, and oncology diseases.

It is not clear characteristic of children in the control group.

Colleagues should include the data of the Ethical approval of the study and declare the consent of the parents for children to participate.  

The number of patients in the study group (n=13) is not sufficient to perform representative statistical analysis.

Colleagues did not conclude on sensitivity and specificity of IP-10 that would increase scientific value of the study.

Conclusions are not clear, although the study itself is comprehensive and analysis was performed on the high level. Still aim and objective of the study are not fully evident.

It is recommended to correct the title of the study.  Diagnostic value of IP-10 in TB has been analyzed. The proposed title – e.g. «Diagnostic value of IP-10 level in children with tuberculosis and other lung diseases»

The results of the study are valuable for TB diagnosis, as currently this problem has not been solved yet. References’ numbers require adjustment in accordance with rules of the journal. The authors should apply the epidemiologic data that is presented in the WHO Report for 2021.

The aim of the study needs re-assessment to make it consistent with design of the study.

Data on the lines 62-77 is not enough clear and may need re-structuring.

Inclusion criteria should be reconsidered: clinical suspicion of TB should not be the inclusion criterion, because TB children often may have no evident clinical symptoms. 

It should be included in exclusion criteria – HIV infection, chronic diseases in exacerbation stage, and oncology diseases.

It is not clear characteristic of children in the control group.

Colleagues should include the data of the Ethical approval of the study and declare the consent of the parents for children to participate.  

The number of patients in the study group (n=13) is not sufficient to perform representative statistical analysis.

Colleagues did not conclude on sensitivity and specificity of IP-10 that would increase scientific value of the study.

Conclusions are not clear, although the study itself is comprehensive and analysis was performed on the high level. Still aim and objective of the study are not fully evident.

Author Response

  1. It is recommended to correct the title of the study. Diagnostic value of IP-10 in TB has been analyzed. The proposed title – e.g. «Diagnostic value of IP-10 level in children with tuberculosis and other lung diseases».

We appreciate your comment and suggestion for the new title, which stresses the importance of the diagnostic value of IP-10 in TB. Due to the fact, that IP-10 in childhood TB has been already studied by others in blood and urine, we would like to emphasize the novelty of our study, which lines in examining IP-10 level in bronchoalveolar lavage fluid. Therefore we would like to suggest another version of the title:

“Diagnostic value of IP-10 level in plasma and bronchoalveolar lavage fluid in children with tuberculosis and other lung diseases”.

  1. The results of the study are valuable for TB diagnosis, as currently this problem has not been solved yet. References’ numbers require adjustment in accordance with rules of the journal. The authors should apply the epidemiologic data that is presented in the WHO Report for 2021.

Thank you for this comment. We are aware of the latest WHO Global TB report results that demonstrated an enormous global drop in the reported number of new TB cases. Close to half of the people ill with TB were missed out in 2020 due to COVID-19 pandemic. Importantly, our study participants were recruited in 2012-2015 when the epidemiological situation was different. We decided to add information on the proportion of children accounting for all TB cases reported annually, as it remained more or less on the same level within the previous years.

  1. The aim of the study needs re-assessment to make it consistent with design of the study.

We are thankful for that comment and agree with the reviewer. We changed it to:

“The aim of the present study was to examine the level of IP-10 in BALF and plasma in children infected with Mycobacterium tuberculosis and children with other respiratory diseases from a low TB-endemic country.”

  1. Data on the lines 62-77 is not enough clear and may need re-structuring.

Thank you for your notice. This is not the part of our manuscript, but probably a part of the guidelines for the authors. We do not know how this got included in the text, as the uploaded version of the manuscript did not include it.

  1. Inclusion criteria should be reconsidered: clinical suspicion of TB should not be the inclusion criterion, because TB children often may have no evident clinical symptoms.

Thank you for paying attention to this issue. We do agree that many paediatric TB patients do not present clinical symptoms of the disease. However, in our study we included only children, who during diagnostic workup had a strong suspicion of TB (history of TB contact/ positive TST and/or IGRA result, and due to this suspicion and clinical presentation were scheduled for a bronchofiberoscopy with BAL. To make it clear we added appropriate information in methods section. Bronchofiberscopy in children is an invasive procedure requiring sedation and general anaesthesia. Therefore, from the ethical point of view, it would be impossible to perform a study using bronchofiberoscopy in children without any clinical manifestation of the disease. Our study was clinical based, and all enrolled children were the patients of our clinic. Only those children who were scheduled for bronchofiberoscopy by their leading physician because of TB suspicion were eligible to participate in the study group. Therefore, we would opt for clinical suspicion of TB remaining inclusion criterion. 

Additionally, to avoid confusion we decided to exclude one patient from the study group, who was referred to our clinic due to a household contact with TB, but was identified as an uninfected TB contact (the boy was followed up for the next six months and remained classified as TB contact). Therefore, we had to recalculate same of the results and introduce minor adjustments not affecting neither results nor conclusions.

  1. It should be included in exclusion criteria – HIV infection, chronic diseases in exacerbation stage, and oncology diseases.

Thank you very much for this remark. We do agree. We used the same criteria in another study, but here we did not provide them in the methods. No children with such conditions were included in the present study. Accordingly, we added these exclusion criteria in the materials and methods section. 

  1. It is not clear characteristic of children in the control group.

We expanded on study and control group characteristics in a new version of Table 1.

  1. Colleagues should include the data of the Ethical approval of the study and declare the consent of the parents for children to participate.

This information is included in the materials and methods section, lines 88-90:

“The study protocol was approved by the Medical University of Warsaw Ethics Committee and informed consent was obtained from the child’s parent or legal guardian before enrollment.”

               An appropriate annotation may be also found in the Back Matter, lines 308-312.

  1. The number of patients in the study group (n=13) is not sufficient to perform representative statistical analysis.
  2. Colleagues did not conclude on sensitivity and specificity of IP-10 that would increase scientific value of the study.

We are aware that a small sample size is the main limitation of our study. Poland is a low – TB-endemic country, and childhood TB is particularly rare. For example, there were 39 new TB cases reported in children aged <15 years in the whole country in 2021. Among children under diagnosis for TB the majority does not need bronchoscopy, which is an invasive procedure requiring sedation and general anaesthesia. While childhood TB in Poland is rare, the need to perform bronchofiberoscopy is even more so. To this study we enrolled all children undergoing bronchofiberoscopy due to TB suspicion who were hospitalized in our department during the study period. Furthermore, within the years following 2015, when the last patient was included, only single patients underwent bronchoscopy as a part of TB diagnostic workup in our clinic. Therefore, we were unable to enrol more children to the study group. In case of other rare diseases results are often presented to the extent that data are available.

Due to the small study group our statistical analysis is limited. As there were only 12  Mycobacterium tuberculosis-infected children (3 with active TB) in our study group, we resigned from setting the cut-off value of IP-10 to discriminate Mycobacterium tuberculosis-infected from uninfected individuals. Neither did we perform sensitivity and specificity analysis. We did, however, demonstrate that IP-10 concentration can be assessed in unstimulated BALF in children. We hope that finding a cut-off value and assessing IP-10 performance characteristics will be a subject of future studies in a larger population.

  1. Conclusions are not clear, although the study itself is comprehensive and analysis was performed on the high level. Still aim and objective of the study are not fully evident.

We corrected the aim of the study section to better reflect the study design. While the diagnostic value of IP-10 has been assessed in children in blood, serum, and urine, we are not aware of a study evaluating its concentration in BALF. Moreover, the majority of studies published to date incorporate in-vitro Mycobacterium tuberculosis-specific antigens to stimulate the production of IP-10, because higher concentrations are easier to detect. The aim of the present study was to investigate whether it is possible to measure IP-10 concentration in children in unstimulated BALF, which we proved. Furthermore, we showed that in children with other lung diseases this concentrations are even higher than in unstimulated plasma.

After the corrections mentioned above we believe that the conclusions are now clear.

Reviewer 2 Report

this was a potentially interesting study however there is a huge problem with the pateitns selections. Specifically, they mention a group of TB infected children. However this group is totally not characterized, as the control group. The control group may be done of complitely different popualtions.

The TB group also only includes only 3 kids with TB, but we dont know any details. Moreover, including in the same group TB and LTBI makes no sense since these two conditions are totally different.

Sorry, but such a study cannot be published.

Best

Author Response

  1. this was a potentially interesting study however there is a huge problem with the patients selections. Specifically, they mention a group of TB infected children. However this group is totally not characterized, as the control group. The control group may be done of completely different populations.

Thank you very much for this comment. We thoroughly reconsidered our study group and introduced several changes. First of all, we added clinical information on patients inclusion and exclusion criteria, explained the process of decision making to perform bronchofiberoscopy with BAL, which was dependant on leading physicians, and provided more information on study group characteristics in the new version of Table 1. To avoid confusion we decided to exclude one patient from the study group, who was referred to our clinic due to a household contact with TB, but was identified as an uninfected TB contact (the boy was followed up for the next six months and remained classified as TB contact). Therefore, we had to recalculate same of the results and introduce minor adjustments not affecting neither results nor conclusions.

  1. The TB group also only includes only 3 kids with TB, but we dont know any details. Moreover, including in the same group TB and LTBI makes no sense since these two conditions are totally different.

We are aware of our study limitations, first of all the small sample size. Our study was conducted in a low-TB endemic country, and all enrolled children were Polish. Childhood TB is rare in Poland. For example, there were 39 new TB cases reported in children aged <15 years in the whole country in 2021. Furthermore, bronchofiberoscopy in children is an invasive procedure requiring sedation and general anaesthesia and the decision to perform it is made individually and with caution. While childhood TB in Poland is rare, the need to perform bronchofiberoscopy is even more so. To this study we enrolled all children undergoing bronchofiberoscopy due to TB suspicion who were hospitalized in our department during the study period. Furthermore, decision to perform bronchofiberoscopy was more frequent when IGRAs were not so available in Poland as they are now. Within the years following 2015, when the last patient was included, only single patients underwent bronchoscopy as a part of TB diagnostic workup in our clinic. Therefore, we were unable to enrol more children to the study group.

When we compared 3 children with active TB with LTBI group we found no significant differences in terms of group characteristics, IP-10 level or BALF cytology. We acknowledge the small sample size. Nevertheless, we assessed IP-10 level in unstimulated BALF and unstimulated plasma, and would rather expect large differences between these two groups first of all in samples stimulated with M. tuberculosis specific antigens. Due to small sample size we were unable to show differences between active TB and LTBI in the present study.  The aim of the present study was to assess if unstimulated BALF can serve as an additional sample for IP-10 measurement. Further studies are needed to evaluate discriminatory properties of IP-10 in BALF.

To better characterize the study and the control group we incorporated additional information in patients and methods section, in Table 1, and in results as mentioned above.

Round 2

Reviewer 1 Report

All my recomendations were takend into account 

Author Response

Once again thank you very much for your comments and suggestions so far.

According to the second reviewers suggestions we added a subanalysis of the study group separating active TB from LTBI. We provided new data in table 2 and figure 2.

Reviewer 2 Report

the paper is improved now.

however I still suggest to separate active tb and LTBI, maybe adding a further subanalyses with new tab and figure, in addition to what has been done so fare

Author Response

Once again thank you very much for your comments and suggestions.

According to your suggestions we added a subanalysis of the study group separating active TB from LTBI. We provided new data in table 2 and figure 2.

Round 3

Reviewer 2 Report

well done, much improved now.

please highlight now in the discussion the differences bcetween active and Ltbi,  but also mention the limited number in the different groups.

last, add in discussion studies showing diagnostics to differentiate active and LTBI, such as:

  • PMID: 30267797
  • DOI: 10.1016/j.jinf.2018.09.011

or b kampmann e biomedicine 2021